# An Analysis of COVID-19 Global Guidelines Published in the Early Phase of the Pandemic for People with Disabilities

**DOI:** 10.3390/ijerph18147710

**Published:** 2021-07-20

**Authors:** Jeong-hyun Kim, Seungbok Lee, Yun-hwan Lee, Jongbae Kim

**Affiliations:** Yonsei Enabling Science and Technology Research Center, College of Health Sciences, Yonsei University, Wonju 26493, Korea; otrehab486@gmail.com (J.-h.K.); supermandrlee@gmail.com (S.L.); enn1210@gmail.com (Y.-h.L.)

**Keywords:** coronavirus, COVID-19, infection, disability, guidelines, review

## Abstract

Purpose: COVID-19 guidelines for persons with disabilities published globally during the early phase of the pandemic by non-governmental organizations and federal agencies were reviewed and analyzed by trends of information provided under various settings. Method: The Google search engine was used by applying the following search terms: COVID-19, Coronavirus 2019, Disability, and Guidelines. Search efforts yielded 514 records from 1 December 2019 to 16 May 2020. The selected 26 guidelines were classified for analysis by organizations (NGOs, non-profit, and governmental institutions), information provided (risks, prevention, and countermeasures), target group (people with disability, service and support providers, and family members), and environmental setting (hospital, community, and home). Results: Government agencies from eight countries published results. Eight of the 26 guidelines were presented by non-governmental organizations, and 18 were not. There were 15 guidelines for individuals with disabilities; seven for service providers, staff, and families providing care; and four addressing both the individuals with a disability and care providers. In terms of appropriate environment and scope, there were 19 guidelines produced for community, government, home, and hospital. The information predominantly presented regarded the prevention of COVID-19 with 22 sources, followed by general information containing risks and response strategies. Conclusion: The majority of the published guidelines focused primarily on the risks and prevention of COVID-19 for people with disabilities. Future procedures should include specific methods in guiding COVID-19 response strategies for the disabled and caregivers who provide essential health services with access to online resources in multiple languages and dialects.

## 1. Introduction

On 11 March 2020, the World Health Organization declared a global pandemic due to the coronavirus (COVID-19) outbreak that resulted in 118,000 infected cases and 4291 deaths across 114 countries [1,2]. In response, individual governing bodies throughout the globe devised strategies and policies to mitigate and prevent the further spread of the deadly virus. The implemented preventive measures included plans for social distancing, restrictions on air travel, and shutting down borders across many countries. For example, in the Republic of Korea (S. Korea), various policies and strategies, such as drive-through test sites, analysis of contact tracing, disinfection of public areas, utilization of digital maps, and community emergency alerts via smart devices, favorably impacted the early efforts to prevent further spread, with expeditious testing of COVID-19 infections [3].

The pandemic has also impacted individuals with disabilities. Historically, this vulnerable population has faced de-prioritization and inequity of health during periods of disaster and public health crisis [4,5]. Cases of infection yielding death among these individuals have been reported in the media. The New York Times reported that 37 out of 46 residents with developmental disabilities cared for by a nonprofit disability services organization were infected during the initial stage of the outbreak, which led to confirmed deaths in many of these marginalized individuals [6,7,8]. In S. Korea, news coverage reported its first death of a severely disabled person following confirmation of COVID-19 infection [9]. In Australia, children with disabilities and their families faced difficulties in obtaining necessary medications. They were pre-empted from accessing urgent healthcare due to a lack of pertinent information and contingent plans for ensuring their safety during this critical period of the outbreak [10].

The advent of the pandemic crisis prompted demographic studies on the global incidence and mortality due to COVID-19. One study (Turk, 2020) reported a higher prevalence of comorbidities among persons with disabilities associated with COVID-19 outcomes, potentially enhancing mortality risk, with more significant risk to those with intellectual and developmental disabilities, especially at younger ages [11]. However, there is sparsely reported data regarding COVID-19 trends in people with varying disabilities [6]. The lack thereof presents a knowledge gap and supports the critical need for increased efforts to investigate the trends in COVID-19 risks, prevalence, countermeasures, and mortality among this population [12].

The first systematically organized COVID-19 guideline of its sort, broadly designed for the disabled population, was released in mid to late March of 2020. The prevailing and unfortunate reality was that the policies containing critical information or recommendations were unavailable at a highly critical period. The strategic implementation of response measures with timely dissemination of essential information was imperative to ensure that this population of marginalized individuals had access to life-saving response measures. The situation was further exacerbated by conflicting recommendations and preventive measures [12,13,14] for this vulnerable community. The pandemic swiftly became a crisis within a crisis for a population who were already challenged by pre-existing comorbidities, economic burdens, physical barriers [15], and social stigmatization, all of which further limits access to essential healthcare for people with disabilities.

Herein, for the purpose of the study, we define the term “guidelines” as the collection of critical information with recommendations provided by government or non-governmental organizations (NGOs) of individual respective countries. The information should communicate specific instructions and contingency plans for receiving essential healthcare, including necessary pharmacotherapeutics and expeditious COVID-19 testing for adults and children with disabilities. The recommendations should address practical methods to preventive measures precisely applicable within their cultural demands and presented in their native language or dialect. These guidelines should provide solutions amid this public disaster, wherein individuals with disability are challenged by social and environmental barriers that inevitably heighten the underlying inequity of health within this population. This study aims to analyze, in qualitative form, the contents of the applicability and limitations of COVID-19 related guidelines for people with disabilities that were presented in the critical early phase of the pandemic, and aims to identify deficiencies in these earlier guidelines to promote focused direction and guidance in future research endeavors that will substantiate their applicability among individuals with disability.

## 2. Methods

### 2.1. Search Strategy and Selection Criteria

We used the Google search engine to identify records, followed by a stepwise process to assess eligibility for inclusion in this review. Keywords with Boolean search terms were used as follows: “COVID-19” or “Coronavirus 2019” and “disability” and “guideline”. Truncations to key search terms were not applied. The search dates were limited from December 2019 through 16 May 2020, taking into consideration the critical escalating period of the COVID-19 spread. Sources that allowed website access to available full guidelines were selected. The selected language was in text format written in English. Photographs and video files for publicity or advertisement were excluded, as were full research articles, since the handful that were available at the time of search did not provide relevant data.

### 2.2. Procedure

The initial 514 records were identified based on our search criteria. Among these, 226 guidelines were classified after excluding sources specific to the elderly and homeless, pregnant women, patients with underlying diseases, and children without disability. Upon review of these 226 records by table of contents, subheadings, information on website links to COVID-19, contact information for human and economic support, updated news on disabled communities within respective countries, statistical data on the outbreak, and frequently asked questions with questions and answers, we obtained 112 data sources. These records were assessed for basic informational contents communicated via poster format with health-related checklists, such as hand washing, mask wearing, social distancing, and other relevant information from central or non-governmental organizations. Duplicate and irrelevant records were all excluded. A total of 26 guidelines were selected and analyzed in the final review (Figure 1).

### 2.3. Data Extraction

The 26 guidelines meeting inclusion criteria were selected after screening for eligibility and included in the final review according to the following data categories: (1) date of issue, (2) organization of origin, (3) targeted subjects, (4) environmental setting, and (5) information provided. In addition, “Setting” was further classified into subcategories: (a) government, (b) hospital, (c) community, and (d) family (home). “Information provided” was also subcategorized into: (a) risks, (b) prevention, and (c) countermeasures. The classifications and subcategories based on data acquisition are summarized in Table 1.

## 3. Results

### 3.1. Classification of Guidelines

Upon classifying data from the selected records by date, there were no specific guidelines for people with disabilities dating from December 2019 to February 2020 in geographical areas where the coronavirus outbreak occurred early. Subsequently, twelve guidelines were published in March of 2020, seven in April, and an additional seven by 16 May 2020. When the records were categorized by the presenting NGOs—the WHO, International Disability Alliance, and UNICEF—eight guidelines were in publication. Government specific agencies, among eight countries, published 18 guidelines. There were 15 guidelines for people with disabilities, seven guidelines for service providers to the disabled, and four guidelines specific for both groups (disabled and providers). When classified according to applicable “Setting” and scope of guidelines, there were eight for government, six for hospitals, seven for family members, and the most publications, 19 in total, were specific for community. In classifying the types of information presented by individual guidelines, there were seven accounts for COVID-19 risks identified for persons with disabilities, six for countermeasures against infection with self-isolation, and 22 specific for the prevention of COVID-19 in general (Figure 2).

### 3.2. Data Comparison of NGO and Country

The COVID-19 guidelines published by two entities—NGO representatives and individual country—were compared according to their target groups, coverage, and type of information provided (Figure 3). Two to three pieces of data in one guideline were found to be duplicates. With respect to target groups, NGOs, and country provided, the highest number of guideline information for the disabled was 16 and nine, respectively; this was followed by guidelines for service agencies, with two and four, respectively. Four guidelines were published by the respective country for both target groups, individuals with disabilities, and service agencies. However, NGOs did not provide any for either group.

In the category for “Setting”, NGOs produced five guidelines for community and three guidelines for government and family. However, they did not publish guidelines specific for hospitals. On the other hand, in category of country, the guidelines for community were the most concentrated with 14 records: five for government and four for family. Six guidelines, which had not been previously provided, were published for the hospital environment by NGOs.

In assessing classifications by the type of information, the NGOs and country presented the most information—six and 26, respectively—regarding preventive measures against the further spread of COVID-19. In addition, a similar number of guidelines have been published by both entities for risks and response measures to actual infection and isolation precautions.

### 3.3. Data Comparison of Target Groups

The target groups—individuals with disability, and service providers (family, caregiver, staff and teacher) or both—were classified into three individual groups for comparison of the respectively presented guidelines (Figure 4). Under “Setting”, guidelines for community provided the predominant information among the three aforementioned groups. In particular, guidelines for the disabled in the community had 14 sources. These included four specifically for the disabled and service providers groups. A mere two to three guidelines specific to government and family were posted. The hospital subcategory for the disabled produced four, and there was only one for service providers.

In the classification according to the type of information provided, the prevention of COVID-19 for the disabled was provided with 17 guidelines (including four for both target groups) and service providers with nine guidelines. On the contrary, sources for COVID-19 risks with its countermeasures for the disabled were relatively scant, with one to three guidelines in total.

## 4. Discussion

Since the global escalation of the coronavirus, studies of COVID-19 trends among persons with disabilities have gradually accumulated throughout the literature [14,42,43]. Across the 26 sources we reviewed, there were deficiencies in providing detailed guidance with specific information on response strategies to individuals with disabilities. The results of our analysis showed that guidelines intended for these individuals focused primarily on prevention that was broadly applicable to the general population. The majority of publications categorized under country were generated by developed areas that presented information limited to and within their own national policies, institutions, and environment. Many were published in English, which presented an obvious limiting factor to those in non-English speaking geographic locations [13]. Thus, appropriate forms of communication (sign language, audio, braille, pictograms, etc.) is indispensable to cover a broader population of people with varying disabilities cross-culturally [44].

Although we observed sparse information on community-dwelling individuals with chronic spinal cord injuries, one study reported that clinicians had difficulty in accurately screening and diagnosing COVID-19 among this population with impaired mobility [42]. Another study showed that the limited or rationed support and aid provided by caregivers had a prominent impact on behavioral changes among individuals with intellectual and developmental disabilities [43]. This was attributed to a lack of education among those who were affected by the heightened risks and consequences of COVID-19 infection [45], whereas the efforts of health education were substantiated as a critical factor in containing further spread [46].

The majority of people with disability who require service care providers are highly dependent on others in performing functional tasks [47,48]. These include basic life activities, such as feeding, toileting, bathing, and transferring in and out of bed. The situation particularly applies to individuals with intellectual and developmental disabilities. Others with visual or mobility impairment and upper or lower limb dysfunction are also dependent, and many are usually unable to function without the aid of caregivers [48]. When a person with a disability is suspected to have been infected, appropriate and prompt intervention must be rendered in a timely manner. Guidelines must present pertinent instructions and be precisely applicable under such circumstances [49].

Healthcare professionals have varying levels of knowledge and clinical experience with patients who have disabilities [50]. Standardized strategies for prevention and input toward COVID-19 response measures are imperative in educating clinicians and caregivers so that the continuity of care is maintained at home, school, and in the community [51]. Federally funded institutions should also ensure that people with disabilities receive proper services provided by the disability workforce [52] during the pandemic.

Many individuals with varying disability are fortunate to receive care from family members who wear the “hats” of caregivers; they play a key role, and their support is vital to ensure the overall wellbeing of these individuals [53,54]. Their caring efforts should not be hampered [55] as basic needs in healthcare and individualized requirements at home, school, and in the community are essential during such an unprecedented time [52]. Consequently, caregivers are at a high risk of infection and are potential carriers, as they shift from one home or location to another, caring for the multitude under many environmental settings. Thus, guidelines must clearly indicate proposed strategies that would minimize risks of infection even within the health service provider industry necessary to protect the health and safety of these individuals [56,57].

There were no guidelines posted for those residing in remote and underdeveloped geographic locations. Unique methods of healthcare and service delivery are required to ensure that vulnerable populations in these areas receive equitable care in a timely manner. Thus, countermeasures and prevention strategies for implementation in remote areas should be considered as a priority. The model proposed by Lakhani (2020) supports prioritization of the vulnerable who face poor access to health services during a public health crisis [53]. Herein, a spatial method framework identified priority areas for essential health services in a time of crisis. Travel times from these areas to medical services were calculated, and spatial analysis identified priority areas with a high percentage of aging adults with disabiy who are confronted by barriers to primary health services. The methods of various healthcare and service delivery, including Lakhani’s proposed methodology [53], are considered as valuable lessons learned for the potential implementation and support of evidence-based COVID-19 response measures for the disabled community amid the pandemic.

The pandemic crisis unavoidably poses a great risk to the global economy; however, most governing bodies and institutions poignantly disregard the aspects of economic burdens in their respective response policies [58]. The situation reiterates the underlying realities of health disparity among the disabled population, and accentuates that the individual rights to healthcare applies to all people, regardless of their social and health status. Thus, the absence of precise countermeasures for the disabled community can be considered as a violation of basic human rights [47]. NGOs and government agencies in each country, based on relevant research, should implement a systematic and strategic plan that aids people with disabilities to respond in a safe and timely manner [44,47,48]. Likewise, healthcare experts and legislators should prepare alternative plans and specific guidelines based on data acquisition from reliable research efforts [59].

Limitations in this study include the limited source of search engines, guidelines printed in English language at the time of our search efforts, and the quantitative synthesis. Review of full-text accessible guidelines generated in multiple languages accompanied by quantitative data will serve to be advantageous. The narrative tends to weigh more toward the population of mobility and intellectual disabilities, while scant information regarding visual and other types of disabilities were limited to a comprehensive qualitative synthesis.

## 5. Conclusions

The public health crisis fueled by the global spread of COVID-19 poses an increased health risk to all people with disabilities and related individuals of varying capacity. Further in-depth and comprehensive research is warranted. The contents of future guidelines should include specific methods in response strategies, including prevention and countermeasures. The dissemination of critical information should be applicable to all individuals with disabilities and their caregivers (family, health service providers, schoolteachers, and clinicians) in multiple languages and made available for those in remote and underdeveloped geographic locations.

## Figures and Tables

**Figure 1 ijerph-18-07710-f001:**
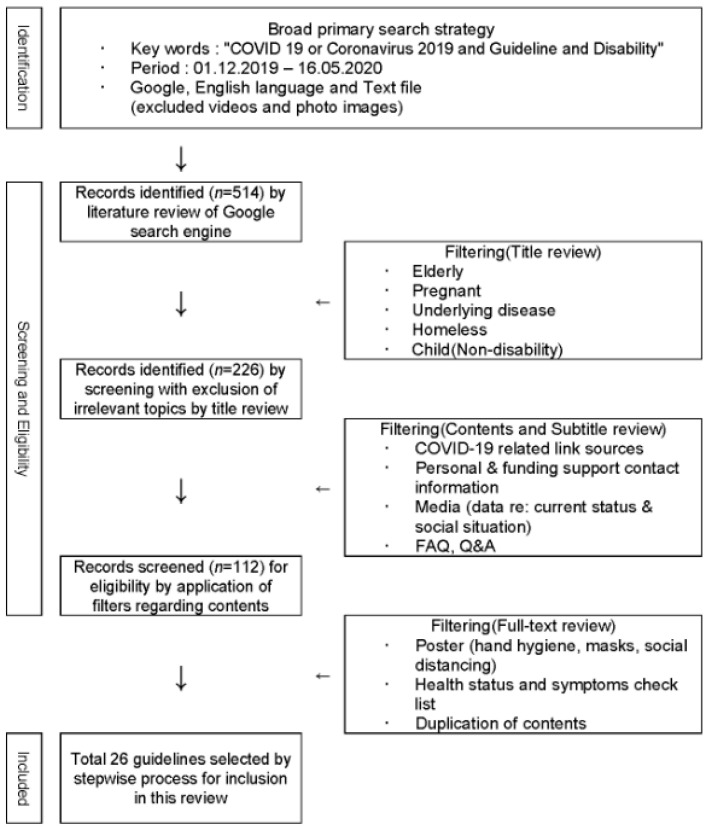
Flow diagram.

**Figure 2 ijerph-18-07710-f002:**
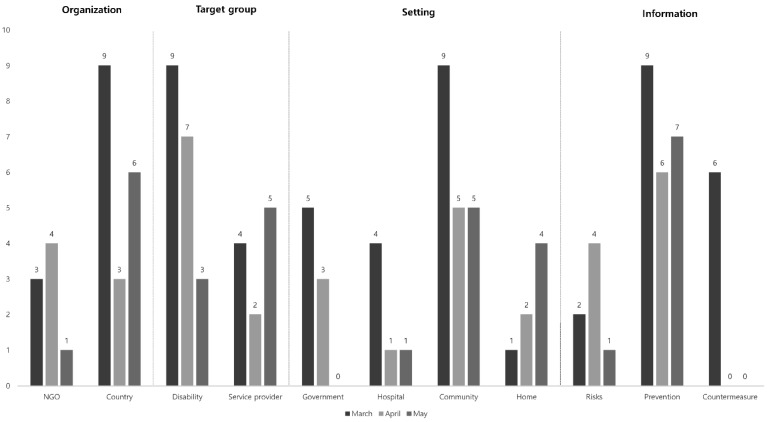
Guidelines published by entities between March through May.

**Figure 3 ijerph-18-07710-f003:**
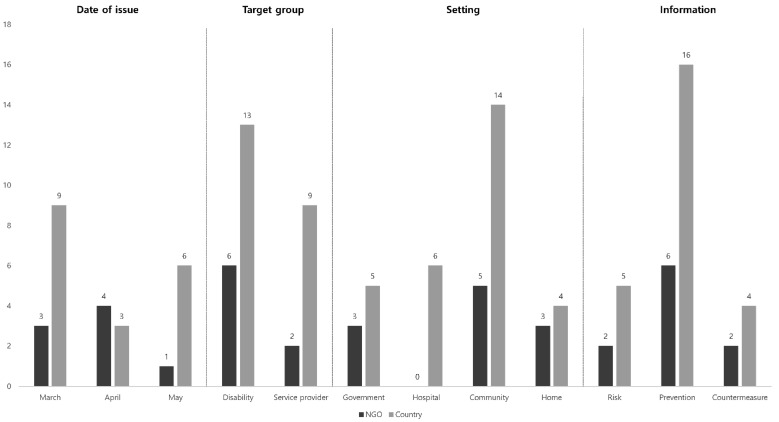
Guidelines published by NGO and country.

**Figure 4 ijerph-18-07710-f004:**
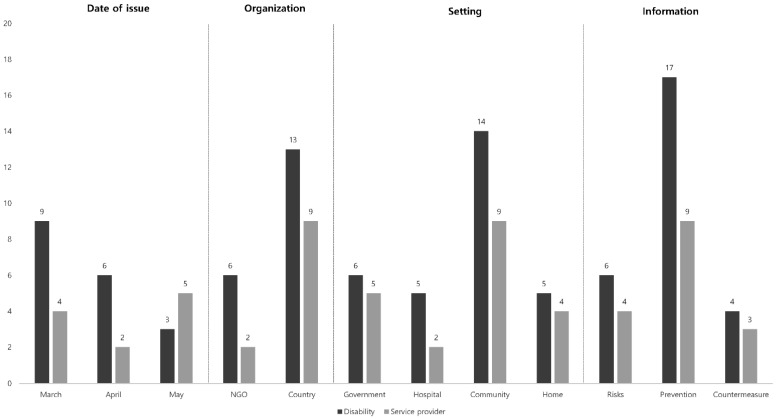
Guidelines applicable to disability and service provider entities.

**Table 1 ijerph-18-07710-t001:** Summary: The early phase of COVID-19 pandemic guidelines for people with disabilities.

Date of Issue	Organization	Title	Target Group	Setting (Environment)				Information Type Provided		
				Government	Hospital	Community	Home (Family)	Risks	Prevention	Countermeasure
13 March 2020	Daegu Solidarity Against Disability Discrimination (Republic of Korea) [16]	Guideline on hospitalization of COVID-19 confirmed PWD	All people with disabilities		√					√
18 March 2020	World Federation of The Deaf/World Association of Sign Language Interpreters (NGO) [17]	Guidelines on Providing Access to Public Health Information in National Sign Languages during the Coronavirus Pandemic	Sign language interpreters or translators	√						√
18 March 2020	Government of South Australia (Australia) [18]	Information on coronavirus (COVID-19) for people with spinal cord injury in SA and NT	People with spinal cord injury			√			√	√
19 March 2020	International Disability Alliance (NGO) [19]	Toward a Disability-Inclusive COVID-19 Response: 10 recommendations from the International Disability Alliance	All people with disabilities			√			√	√
19 March 2020	Department of Health and Social Care (UK) [20]	COVID-19 Hospital Discharge Service Requirements	Community health services and social staff	√	√	√				√
23 March 2020	Veterans Health Administration—Office of Emergency Management (USA) [21]	COVID-19 Response Plan	Veterans and staff personnel	√	√	√		√	√	√
24 March 2020	Washington Office of Superintendent of Public Instruction (USA) [22]	Questions and Answers: Provision of Services to Students with Disabilities During School Facility Closures for COVID-19	Students with disabilities			√			√	
25 March 2020	Ministry of Home Affairs (India) [23]	Comprehensive disability inclusive guidelines for protection and safety of persons with disabilities (Divyangjan) during COVID 19	All people with disabilities	√				√	√	
26 March 2020	World Health Organization (NGO) [24]	Disability considerations during the COVID-19 outbreak	All people with disabilities	√		√	√		√	
27 March 2020	Faculty of Psychiatry of Intellectual Disability (UK) [25]	Faculty of Psychiatry of Intellectual Disability COVID-19 and People with Intellectual Disability	People with intellectual disabilities		√	√			√	
30 March 2020	Queensland Spinal Cord Injuries Service (Australia) [26]	Planning for disruption to supports and services because of COVID-19 advice for people with spinal cord injury	People with spinal cord injury			√			√	
31 March 2020	Disability Services Consulting (Australia) [27]	COVID-19 in Disability Accommodation	Supported independent living providers			√			√	
03 April 2020	The British Association of Spinal Cord Injury Specialists (UK) [28]	Basics guidance of management of spinal cord injury patients during coronavirus (COVID-19) pandemic	People with spinal cord injury		√				√	
08 April 2020	UNICEF (NGO) [29]	COVID-19 response: Considerations for Children and Adults with Disabilities	Children and adults with disabilities			√			√	
09 April 2020	Global Protection Cluster: Syria Protection Cluster (Turkey) (NGO) [30]	A disability-inclusive COVID-19 response	All people with disabilities			√	√	√		
09 April 2020	Australian Government: Department of Health (Australia) [31]	Management and operational plan for people with disability: Australian Health Sector Emergency Response Plan for Novel Coronavirus	All people with disabilities, families, caregivers, support workers, disability and healthcare sectors	√		√	√	√	√	
15 April 2020	Inclusion Ireland National Association for People with an Intellectual Disability (Ireland) [32]	COVID-19 and Intellectual Disability: Supporting people with intellectual disabilities and their families	People with intellectual disabilities and their families	√		√		√	√	
28 April 2020	Save the Children (NGO) [33]	Tip Sheet for Disability Inclusion during COVID-19 child protection, education, health, nutrition, WASH	Children with disabilities			√		√	√	
29 April 2020	United Nations (NGO) [34]	COVID-19 and the rights of persons with disabilities: Guidance	All people with disabilities	√					√	
05 May 2020	UNESCO New Delhi Cluster Office (NGO) [35]	Life in the Times of COVID-19: A guide for parents of children with disabilities	Parents of children with disabilities				√		√	
05 May 2020	NJ Department of HumanServices Division of Developmental Disabilities (USA) [36]	COVID-19 Guidance for Individuals and Families of Individuals with Intellectual and Developmental Disabilities	People with intellectual disabilities and their families			√	√		√	
11 May 2020	Ministry of Health (New Zealand) [37]	Mental health resources for disability support service providers	Disability support service providers			√			√	
12 May 2020	Ministry of Health (New Zealand) [38]	Alert Level 2 guidance for disability support service providers	Disability support service provider			√		√	√	
12 May 2020	Ministry of Health (New Zealand) [39]	Clinical guidance for responding to patients with an intellectual (learning) disability during COVID-19 in Aotearoa, New Zealand	People with intellectual disabilities		√			√		
12 May 2020	Beneficial Designs Inc. Minden, NV (USA) [40]	Attention: Wheelchair and Assistive Technology Users Precautions for COVID-19	Wheelchair and assistive technology users			√	√		√	
15 May 2020	Ministry of Health (New Zealand) [41]	Guidelines for personal protective equipment (PPE) disability support care workers who work in clients’ homes	Support and care workers providing support in clients’ homes			√	√		√	

## Data Availability

This study did not generate any new data; and there was no quantitative synthesis. The tables and figures in the article shows data extracted from published resources for illustrative purposes and is summarized by a flow diagram and bar graphs.

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
