# Peer review of "An Analysis of COVID-19 Global Guidelines Published in the Early Phase of the Pandemic for People with Disabilities"

_ijerph, 2021, doi:10.3390/ijerph18147710_

Round 1

Reviewer 1 Report

The work presented is of enormous interest and relevance. One of the groups most affected by the COVID-19 pandemic has been people with functional diversity. Therefore, it is worthwhile to review how the recommendation guidelines have been constructed in order to consider how to learn and improve. 

Below we present a series of aspects that could improve the manuscript presented:
- In the introduction, it would be interesting for the authors to delve deeper into how the pandemic has affected people with disabilities in order to better contextualize the work done. It now falls a bit short. 
- In the last paragraph of the introduction, methodological aspects are mixed with the objective. It would be interesting to clearly differentiate the objective from other issues.
- In methods, it would be appropriate for the authors to first explain the type of study they are going to carry out and the methodology chosen, justifying it. 
- Likewise, it would be relevant that in the discussion the authors go more deeply into the lessons learned in order to improve the guidelines that can be carried out in the future. 

Reviewer 2 Report

Thank you for submitting this relevant and interesting paper on an important topic for people with disability managing the COVID pandemic. The aim of your research and the review of published guidelines is an appropriate way of gaining some insights into this issue. However, your introduction does not adequately set up the paper for a consideration of the issues for people with disability in the pandemic context and spends too much time on general information about the pandemic. The method is reasonably described, but I would like to see more rationale provided for the exclusion of guides in different formats as this achieves the aim of reaching a broader group including people with vision impairment of cognitive impairment which is an important consideration. I am also not clear why you excluded research papers. The data extraction table is adequate, but would be enhanced by further narrative content in the Information type columns. The results section is inadequate in my view, providing surface level insights from the documents, based on a numerical presence or absence of features of the documents. This section needs to include narrative descriptions of the content included in the documents, potentially in a thematic analysis to provide synthesis. The discussion section does not clearly link with the results presented and provides insights from the broader literature about issues for people with disability in a pandemic context that would be better placed in the introduction to firmly establish the context. There are some references to the content type of the included documents that is helpful and relevant, but difficult to ascertain the interpretations and conclusions when there is no detail provided in the results section to support these points. The conclusions are not supported by the data presented, and whilst they are very reasonable and helpful need to be linked with content from the results. This paper requires significant work to expand the results section and restructure the discussion and introduction. It is an important topic area and would be of interest to a wider audience if these issues were addressed. 

Round 2

Reviewer 1 Report

The modifications made by the authors satisfy the requirements of this reviewer.